# Effect of Conducting, Semi-Conducting and Insulating Nanoparticles on AC Breakdown Voltage and Partial Discharge Activity of Synthetic Ester: A Statistical Analysis

**DOI:** 10.3390/nano12122105

**Published:** 2022-06-19

**Authors:** Hocine Khelifa, Abderrahmane Beroual, Eric Vagnon

**Affiliations:** Ecole Centrale de Lyon, University of Lyon, Ampère CNRS UMR5005, 69130 Ecully, France; abderrahmane.beroual@ec-lyon.fr (A.B.); eric.vagnon@ec-lyon.fr (E.V.)

**Keywords:** nanofluids, synthetic ester, oxide nanoparticles, AC breakdown voltage, partial discharges, statistical analysis, extreme value distribution, normal and Weibull distributions

## Abstract

This paper is aimed at studying the influence of conducting (Fe_3_O_4_), semi-conductive (ZnO), and insulating (ZrO_2_, SiO_2_, and Al_2_O_3_) nanoparticles (NPs) at various concentrations on the AC dielectric strength of MIDEL 7131 synthetic ester (SE) and partial discharges activity. First, a detailed and improved procedure for preparing nanofluids (NFs) in five concentrations ranging from 0.1 g/L to 0.5 g/L is presented, including high-speed agitation and ultrasonication. Then, the long-term stability is checked based on zeta potential analysis. After preparing and characterizing the NF samples, the following step is to measure their AC breakdown voltage (BDV). Due to the limitation of the high voltage supply (Baur system), the tests are performed according to IEC 60156 standard (2.5 mm gap distance) only with ZnO, ZrO_2_, and SiO_2_ NPs, and for comparison, tests are executed for all considered NPs with an electrodes gap of 2 mm. It is shown that the addition of Fe_3_O_4_ (20 nm), ZnO (25 nm), ZrO_2_ (20–30 nm), SiO_2_ (10–20 nm), Al_2_O_3_ (20–30 nm), and Al_2_O_3_ (50 nm) NPs improves the dielectric strength of synthetic ester upon an optimal concentration which gives the highest AC BDV. SiO_2_ (10–20 nm) and Al_2_O_3_ (20–30 nm) manifest their best improvement at 0.3 g/L, while for the other NFs, the best improvement is observed at 0.4 g/L. Further, the Anderson–Darling goodness-of-fit test is performed on the experimental data to check their conformity with the Extreme value (EV), normal, and Weibull distributions; the normal and EV fit curves are plotted and used to evaluate the breakdown voltages at probabilities of 1%, 10%, and 50%. It is shown that the AC breakdown voltage outcomes for most investigated nanofluids mostly obey the three EV, normal, and Weibull distributions. Then, the best combinations (nature of NP and optimal concentration), namely Fe_3_O_4_ (20 nm, 0.4 g/L), Al_2_O_3_ (20–30 nm, 0.3 g/L), and Al_2_O_3_ (50 nm, 0.4 g/L) NPs, that highly enhance the AC BDV of SE are chosen for a partial discharge activity investigation and comparison with pure SE. It is shown that the addition of those NPs significantly reduces the activity of partial discharges compared to pure SE.

## 1. Introduction

Nowadays, nanotechnology has emerged as one of the most exciting and advancing areas in science and engineering. Most academic centers and industries around the globe have been occupied in focusing on nanoscale research as a part of miniaturization/proficiency in devices. As an indication, biological and medical communities exploit the properties of nanomaterials for a variety of applications under the terms nanobiology and nanomedicine [1,2]. The size of biological structures or molecules is quite close to those materials. Therefore, they could add functions to those structures/molecules. This immense integration allows reliable diagnostic or rapid drug administration tools [3,4]. Moreover, electronic chips or integrated circuits are already nano-scaled component-based and extensively use nanotechnology [5]. Thanks to the emergence of nanotechnology in energy conversation, today’s solar panels yield twice as much energy [5,6]. 

An innovative concept termed “Nanofluids” (NFs) in science and engineering refers to the exploitation of the unique properties of nanoparticles for various applications. Chen et Eastman. [7] introduced it in the late twentieth century. The nanoparticles (NPs) are homogeneously mixed up within a base liquid with a proper technic. A typical example of a nanofluid is the iron oxide (Fe_3_O_4_) nanoparticles dispersed in a host liquid, such as water. In the beginning, it represented the best belief for heat transfer enhancement [7]; this was proven so far, and more recently, the attention has been paid to how better the dielectrics liquids would be in the presence of these particles for insulation applications and how they affect their properties [8,9,10,11,12,13,14,15,16]. Among these properties, the breakdown voltage (BDV) was considered as the essential property to be taken into consideration. Thus, recent years have seen a rise in the number of related publications. 

Olmo et al. [17] reported that the addition of TiO_2_ NPs (10–20 nm) improves the AC breakdown voltage of natural ester (NE) by 33.2% at a concentration of 0.5 g/L, while with the other concentrations, the enhancements are 25.8%, 30.4%, 21.6%, and 7.6% for the concentration of 0.1, 0.2, 0.7, and 1 g/L, respectively. Peppas et al. [13] and Khaled and Beroual [15,16] found a similar tendency with Fe_3_O_4_, Al_2_O_3_, and SiO_2_ nanofluids at different concentrations (0.05 to 0.5 g/L). With equal spherical particle sizes (50 nm), Khaled and Beroual [16] investigated the effect of conductive (Fe_3_O_4_) and insulation nanoparticles (Al_2_O_3_ and SiO_2_) on the breakdown voltage of synthetic ester (SE)-based nanofluids. They found that SE-based NF with Fe_3_O_4_ enhances the BDV about 48% against 25% and 32% with Al_2_O_3_ and SiO_2_, respectively, compared to the host liquid. They reported that the insulation nanoparticles give somewhat the same improvement; conductive nanoparticles provide the best enhancement. Those results put forward the possible implications of electrical conductivity on the breakdown mechanisms. Hussain et al. [8] examined the effect of adding magnetic NPs, namely iron oxide (Fe_3_O_4_), cobalt oxide (Co_3_O_4_), and iron phosphide (Fe_3_P), on the insulation performances of synthetic and natural esters. They show that these NPs enhance the dielectric strength of both types of liquids. Furthermore, the Fe_3_P NFs show the best improvements at an optimal concentration of 0.02 g/L. Those enhancements are 20.2% and 31.4% for SE and NE-based NFs, respectively. 

Hwang et al. [11] explained the AC BDV improvement via electron trapping as a possible mechanism based on the relaxation time constant (τ_r_). This mechanism nicely explains the enhancement of base liquid’s insulating performances with addressed conductive NPs (Fe_3_O_4_), but it fails to explain the performance improvement of other NPs, even conductive ones [18]. Furthermore, Sima et al. [10] examined the effect of the conductivity and permittivity of conductive (Fe_3_O_4_), semi-conductive (TiO_2_), and insulating (Al_2_O_3_) NPs on transformer oil performances. The ionization models of transformer oil-based NFs were developed. They concluded that NPs whose conductivity or permittivity mismatch those of the base liquid increase the saturation charges on their interface, which slows down the evolution of the streamers, thus enhancing the dielectric strength of NFs. 

A. Beroual and H. Duzkaya [19] examined the variation of AC and lightning impulse (LI) breakdown voltages of natural ester enriched with fullerene (C_60_) nanoparticles for concentrations ranging from 0.05 to 0.4 g/L. They observed that the AC breakdown voltage performances of 0.3 and 0.4 g/L C_60_ nanofluids are increased compared to natural ester. The increase rate is 5.1% and 7.8%, respectively. A concentration of 0.1 g/L C_60_ has 8.2% better performance than natural ester at LI breakdown voltages. Recently, Khelifa et al. [20] investigated the AC breakdown voltage of a synthetic ester MIDEL 7131-based NFs with graphene (Gr) and fullerene (C_60_). They found that adding Gr or C_60_ at different concentrations enhances the BDV value, and optimal concentration between 0.3 and 0.4 g/L gives the best breakdown voltage value. The BDV enhancement is more important with SE-based NFs with C_60_ than with Gr for the electrode gaps up to 0.5 mm, with the electrodes being spherical and 12.5 mm in diameter. Beyond this gap, the improvements are more important with Gr than with C_60_. The best improvements for a 2 mm gap are about 12.67% and 16.64%, with C_60_ (0.4 g/L) and Gr (0.3 g/L), respectively. It was reported from the literature that most of the published works address the performances of breakdown voltages of transformer oil-based NFs; the partial discharge (PD) activity on those NFs is less well-studied in the literature. 

Jin et al. [21,22] investigated the PD activity in mineral oil and mineral oil-based NFs with Fullerene (C_60_) and silica (SiO_2_) NPs at 0.01 wt.% (~0.1 g/L). An enhancement of the PD inception voltage (PDIV) by 20% with SiO_2_ and by 10% with C_60_ was reported. They attributed this enhancement to the positive effect of hydrophilic silica NPs, which reduce the moisture content, thus reducing the PD activity. Atiya et al. [23] compared the PD activity of mineral oil and mineral oil-based Al_2_O_3_ NFs impregned pressboards. They reported that Al_2_O_3_ NFs impregned pressboard show the highest PDIV. Furthermore, Atiya et al. [24] investigated the effect of the type of NPs (TiO_2_ and Al_2_O_3_) and the thickness of electrical double layers (EDL) around the NPs; they were the first to explain such difference in PD activity by the EDL around NPs [24]. They concluded that the smaller the EDL, the lower the PD activity. Recently, Khelifa et al. [20] and Koutras et al. [25] reported that the addition of carbonic (C_60_) and silicon carbide (SiC) NPs enhances the PD resistivity of pure synthetic ester (MIDEL 7131) and natural ester (FR3), respectively. 

This paper is aimed at studying the influence of conducting (Fe_3_O_4_, 20 nm), semi-conductive (ZnO, 25 nm), and insulating (ZrO_2_, 20–30 nm, SiO_2_, 10–20 nm, and Al_2_O_3_, 20–30 and 50 nm) nanoparticles (NPs) at various concentrations on the AC dielectric strength of MIDEL 7131 synthetic ester. First, Section 2 presents a detailed and improved procedure for preparing NFs in five concentrations ranging from 0.1 g/L to 0.5 g/L, including high-speed agitation and ultrasonication; the long-term stability checking technic based on zeta potential analysis is then addressed. After dealing with the preparation and characterization of NF samples, the following step presents the AC BDV and PD measurements methodology. Next, section three is devoted to the results of long-term stability, AC BDV, and PD activity. The conformity of experimental outcomes with normal, Weibull, and Extreme Values statistic distributions laws are also performed, and the mean and breakdown voltages at different risk levels were evaluated. Finally, the fourth section discusses the results and the possible physicochemical processes behind the observed improvement of AC BDV and PD resistivity. 

## 2. Materials and Methods

### 2.1. Materials

The used base liquid is the synthetic ester MIDEL 7131 provided by M&I Materials, Manchester, UK. ZrO_2_, SiO_2_, and Al_2_O_3_ NPs were supplied by SkySpring Nanomaterials (SS Nano, Houston, TX, USA), while the ZnO NPs were supplied by PlasmaChem GmbH, Berlin, Germany. The Fe_3_O_4_ NPs and oleic acid (*cis*-9-Octadecenoic acid) were supplied by Sigma-Aldrich (St. Louis, MI, USA). Table 1 and Table 2 show the physicochemical properties of MIDEL 7131 [26] and the characteristic of NPs, respectively. Some properties were not provided by the suppliers; we indicated those taken from the literature, which are marked with an asterisk [10]. A high-speed rotor-stator disperser (Misceo, 250 F, Mortagne-sur-Sèvre, France) and ultrasonic liquid processors device (Sonics, VCX 500, Newtown, CT, USA) were used to prepare NFs. 

### 2.2. Samples Preparation

The two-step method was executed for preparing the NFs samples, as depicted in Figure 1. The base liquid cannot be used in its initial state, and it needs to be purified; the purification was made using a micro-membrane filter and a vacuum pump to remove impurities. Oleic acid (used as a surfactant) was then added, and the mixture was stirred for five minutes using the high-speed rotor-stator mixer at 13,000 rpm. The mass ratio of oleic acid (OA) to base liquid is 0.75 wt.%. This concentration was based on previous work within the research team [20]; the evolution of zeta potential versus concentration of OA has been examined, and the 0.75 wt.% gives the best compromise. Next, the desired weight of powder NPs was dispersed within the base liquid, and the mixture is agitated for 20 min. Five concentrations are considered, ranging from 0.1 to 0.5 g/L (0.01 to 0.05 wt.%). Finally, the samples of NFs are subjected to an ultrasound agitation for two hours to uniformize the mixture and reach a stable colloid. The Ultrasonic Liquid Processors device (500 W, 20 kHz) with 25 mm low-intensity solid probe operates in a pulsed mode (i.e., 10 s of operation and 5 s of rest) with an amplitude set of 60%. After every 30 min, the ultrasonic equipment is rested for 5 min to avoid overheating NF samples and extend the equipment lifetime, especially the solid probe. Note that the volume of the prepared NFs is 400 mL for each concentration and type of NPs.

### 2.3. Stability of Nanofluids

The stability was considered to be the most significant issue facing NFs. Unfortunately, this has proven to be a severe impediment to the widespread usage of those liquids, especially for those applications that need a considerable volume, as the case for power transformers. Thus, particular attention should be paid to this step for a proper outcome concerning colloid stability. Many techniques for checking the stability were reported in the literature, including zeta potential (ζ-potential) analysis [27]. The ζ-potential analysis appears to be the most efficient and less time-consuming method to check the stability of NFs [27,28]. The stability depends on the ζ-potential value; a schematic presentation is depicted in Figure 2. For the absolute values between 0 and 10 mV, the NF is considered unstable, and action is needed to overcome this.

In contrast, for the absolute values higher than 30 mV, the NFs are considered highly stable; between the two ranges, the NFs are stable [28]. The hydrodynamic diameter and ζ-potential measurements were performed thrice using a Zetasizer Nano (ZS) instrument (Malvern, UK). As a supplement to zeta potential measurements, the Zetasizer Nano provides the electrical conductivity of NF samples. Therefore, the spectral absorbance of each sample was checked before performing the measurement. All samples whose spectral absorbance is less than 100 could perform the zeta potential analysis as explained in a previous work [20]. Hence, only three concentrations (0.1, 0.2, and 0.3 g/L) were considered according to the absorbance measurement. 

### 2.4. Procedure for Breakdown Voltage Measurement

The breakdown voltage measurements have been performed in compliance with IEC 60156 standard method using a commercially available BAUR system. The test bench consists of an oil tank with a capacity of 400 mL, an electrodes system with adjustable gap distance, and a high voltage generator that can reach 100 kV RMS (50 Hz). According to IEC 60156 [29], the BDV test is performed in a sphere-sphere electrode configuration of 12.5 mm diameter, with a spacing of 2.5 mm. The voltage is applied continuously with an increment of 2 kV/s until breakdown occurs. 

Fe_3_O_4_ and Al_2_O_3_ NFs show superior dielectric strength, exceeding the oil tester limitation (100 kV); for that reason, a reduced electrode gap to 2 mm is considered to compare the breakdown voltage of the six NPs. In addition, the BDV test is carried out for NFs in which the breakdown occurs for an electrodes gap of 2.5 mm (i.e., ZnO, ZrO_2_, and SiO_2_). Three series of six measurements have been performed, giving 18 points considered sufficient for the statistical analysis [13,16]. Next, the conformity of AC BDV data for 2 mm and 2.5 mm electrode gaps to the extreme value (EV), Weibull, and normal distributions were analyzed using Anderson–Darling statistics. The EV distribution is rarely used to analyze AC-BDV data [13], unlike the widely used Weibull and normal distributions. Finally, the voltages corresponding to 1%, 10%, and 50% risk levels were determined using the normal and EV distributions. 

### 2.5. Partial Discharges Measurement under AC 50 Hz Stress

The partial discharges (PDs) activity in both SE and the three SE-based NFs that give the best improvement in AC BDV tests are conducted in compliance with IEC 60270 standard method; in this case, Fe_3_O_4_ (20 nm) and Al_2_O_3_ (50 nm) at optimal concentration 0.4 g/L and Al_2_O_3_ (20–30 nm) at optimal concentration 0.3 g/L. An industrial Omicron PDs system detection was used for this purpose. The PDs test is performed in a needle-plane electrode configuration; the gap between the two electrodes is 5 mm. The tip radius of curvature is 10 μm, while the plane electrode has a 35 mm diameter. The applied voltage is varied, and its RMS value follows the profile depicted in Figure 3; the voltage rises and falls with a speed of 1 kV/s, 13 kV as the maximum value on the plateau, maintained for 32 s, and 5 s of rest is respected between two successive tests from the same series. This voltage profile is repeated five times for each sample, which underwent five PDs tests for each liquid. So, the collected values present the average of five measurements. 

For the comparison and quantification of PD activity, we are interested in the PD inception voltage (PDIV), PD extinction voltage (PDEV) during raise and falling times, respectively, and average charge (Q_avg_), peak charge (Q_peak_), and number of PDs (NPDs/s) during the voltage plateau (at 13 kV RMS). It is about the average charge and the number of PDs per second (NPD/s) during the 32 s, while Q_peak_ is the highest charge recorded in the same interval. In addition, the phase-resolved PDs pattern is also compared and plotted for the four liquids.

## 3. Results

### 3.1. Stability of Nanofluids

As mentioned above, the hydrodynamic diameter and zeta potential measurements are performed on the NFs samples. However, hydrodynamic diameter analysis does not give an affirmative indication of the stability of NFs. For that reason, one concentration is considered in this analysis for the six NFs to show the dispersion behavior and look at the agglomerations/clusters present in the liquid. Figure 4 shows the size distributions of different SE-based NFs for a specified concentration (0.1 g/L). For all NF samples, it was noticed that maximum intensities reveal sizes higher than the declared ones by the supplier (Figure 4b). Since the observed/measured diameter using Dynamic Light Scattering (DLS) does not exclusively consider the particle size, unlike to NPs size measurement by microscopy technics (SEM and TEM); oleic acid envelopes the NP, possibly leading to an overestimation (an increase of the size), thus to larger sizes than expected. [30]. Furthermore, from what the suppliers claim, Fe_3_O_4_ (20 nm), ZnO (25 nm), and Al_2_O_3_ (50 nm) NPs should present a size distribution with a high peak around a specific diameter (theoretically around the declared sizes), and ZrO_2_ (20–30 nm), SiO_2_ NPs (10–20 nm), and Al_2_O_3_ (20–30 nm) should be much larger (large variation sizes with a smaller peak), while only ZrO_2_ (20–30 nm) and Al_2_O_3_ (20–30 nm) NPs fit this description.

For the zeta potential analysis, the measurement is performed for the three concentrations in which the absorbance values are below 100, which indicates the test feasibility. Table 3 gives a summary of ζ-potential and electrical conductivity results for three concentrations taken one week after the preparation. According to the results, SiO_2_ (10–20 nm) NF is highly stable, while the other NFs are stable. We could speculate this to the size difference (contact surface); SiO_2_ (10–20 nm) is the smaller NPs, which should provide a more significant contact surface NPs/liquid than the other NPs. 

Note that after three weeks, we did not observe sedimentation. However, if the zeta potential indicates that NFs remain stable, this cannot be a guarantee of stability for several years and the lifetime of the transformer.

Moreover, the electrical conductivity of ZrO_2_ (20–30 nm), SiO_2_ (10–20 nm), and Al_2_O_3_ (20–30 and 50 nm) NFs decrease with concentration, while it increases with Fe_3_O_4_ (20 nm) and ZnO (25 nm) NFs. This is because ZrO_2_, SiO_2_, and Al_2_O_3_ are insulation NPs, while ZnO is semi-conductive NPs, and Fe_3_O_4_ is conductive NPs. Those fundamental differences may justify the observed differences.

### 3.2. AC Breakdown Voltage Test for 2 mm Electrodes Gap

As explained in the previous section, the breakdown does not occur at a 2.5 mm electrode gap with Fe_3_O_4_ (20 nm) and Al_2_O_3_ (20–30 and 50 nm) NFs; the AC BDV with a 2 mm electrode gap distance is also considered to compare the six NPs under the same experimental conditions, which is the purpose of this subsection. Figure 5a–f shows the mean and max/min AC BDV for Synthetic Ester and Fe_3_O_4_ (20 nm), ZnO (25 nm), ZrO_2_ (20–30 nm), SiO_2_ (10–20 nm), and Al_2_O_3_ (20–30 and 50 nm) NFs at different concentrations, respectively. Whatever the type and concentration of the used NPs, the enhancement of the BDV in NFs was remarkable compared to the pure SE. In addition, adding NPs reduces the standard deviation of AC BDV which mainly increases the slope in the statistical analysis, thence enhancing the BDV at low-risk levels. Furthermore, the type and concentrations of NPs play a significant role in increment percentage. Synthetic ester-based ZnO (25 nm, ZrO_2_ (20–30 nm), and SiO_2_ (10–20 nm) NPs manifested the best improvements of around 20% compared to pure SE, while the best improvements with Fe_3_O_4_ (20 nm) and Al_2_O_3_ (20–30 and 50 nm) NFs NPs were between 37% and 44% concerning pure SE. Those improvements were compared and plotted, as depicted in Figure 6. It was noted that all NFs samples reveal an optimal concentration of around 0.3 and 0.4 g/L. 

Note that with the optimal concentration of SiO_2_ (10–20 nm) and Al_2_O_3_ (20–30 nm) NPs that is 0.3 g/L, the BDV is improved by 20.30% and 44.12%, respectively, as shown in Figure 5d,e. Up to 0.3 g/L, a slight lowering for the same Al_2_O_3_ NPs of 50 nm is remarked compared to the smaller Al_2_O_3_ of 20 nm NPs; a reversed tendency is observed beyond 0.4 g/L. With 0.4 g/L Al_2_O_3_ (50 nm), the enhancement is 42.13% compared to pure SE (Figure 5f). Khaled and Beroual have also carried out conceptually similar work [15,16] in which Al_2_O_3_ (13 and 50 nm) NPs dispersed within mineral oil and synthetic ester, where they showed a similar tendency: the smallest particles provide a lower optimal concentration. With Fe_3_O_4_ (20 nm), ZnO (25 nm), and ZrO_2_ (20–30 nm) NPs, the best enhancements are about 37.70%, 19.38%, and 21.37%, respectively, for a concentration of 0.4 g/L, as shown in Figure 5a–c. Fe_3_O_4_ (20 nm) and Al_2_O_3_ (20–30 nm and 50 nm) NFs show the best performances according to AC BDV; their best improvements are greater than 35% for all cases.

### 3.3. AC Breakdown Voltage Test for 2.5 mm Electrode Gap (IEC 60156)

Following the IEC 60156 standard (2.5 mm electrode gap), among the six tested NPs, the breakdown occurs only with ZnO (25 nm), ZrO_2_ (20–30 nm), and SiO_2_ (10–20 nm) NFs. Therefore, only the three NPs will be addressed in the following section. Figure 7, Figure 8 and Figure 9 give the mean and max/min AC BDV for synthetic ester-based ZnO (25 nm), ZrO_2_ (20–30 nm), and SiO_2_ (10–20 nm) NFs at different concentrations, respectively. It was noted that the best enhancements of SE-based ZnO (25 nm) and ZrO_2_ (20–30 nm) NFs, are about 14.28% and 11.13%, respectively, for a concentration of 0.4 g/L, as shown in Figure 7 and Figure 8, while for SE-based SiO_2_ (10–20 nm) NFs, the improvement reaches the highest BDV value for a concentration of 0.3 g/L (Figure 9); this presents a 12.83% of improvement against pure SE. Finally, the improvements were compared and plotted for each concentration, as depicted in Figure 10. 

### 3.4. Statistical Analysis of AC Breakdown Voltage Data 

Extreme value (EV), normal, and Weibull distributions are used to analyze and test the conformity of breakdown voltage outcomes. Contrary to the popular statistical laws (i.e., Weibull and normal) [16,20,25,31], EV has been rarely considered to adjust the experimental AC BDV outcomes [13]. EV distribution combines Gumbel, Frechet, and Weibull distribution [13]; hence probability fit curves of the experimental results are plotted using EV and normal distribution. Those latter are used then to estimate the voltage at specific risk levels. Therefore, the most crucial BDV levels should be estimated at 1%, 10%, and 50% risk levels. Before estimating those voltages, a goodness-of-fit test should be performed to check if the data came from a population with a specific distribution. If the distribution obeys the experimental data, the voltages could be estimated, and the results thence present a good estimation.

Nevertheless, the conformity of the experimental data was investigated using the Anderson–Darling test. The Anderson–Darling (AD) statistic is a goodness-of-fit test mainly used to decide whether a sample of size *n* is drawn from a specified distribution, most commonly whether the sample data is drawn from a normal distribution. This test has been successfully extended to the other distributions [32]. Anderson–Darling goodness-of-fit is performed to check if the experimental data comes from EV, normal, and Weibull distribution. The conformity is then decided according to the *p*-value, depending on the AD value [32,33]. Based on the statistics, if the *p*-value (probability value) is higher than the significance level, alpha (α), there is enough evidence to accept the hypothesis that the data come from a specific distribution. The *p*-values of 0.05 were considered statistically significant [13,16,33], and from the AD test, the *p*-value for the EV, normal, and Weibull distributions were calculated and compared to the significance level. 

#### 3.4.1. Statistical Analysis of AC Breakdown Voltage Outcomes for 2 mm Electrodes Gap

The Anderson–Darling goodness-of-fit was performed on experimental data of the six NFs at different concentrations for the 2 mm electrode gap in which the breakdown occurs at this gap distance; the results are shown in Table 4. It was noticed that the *p*-value is higher than the significance level for most cases, and therefore, the experimental data obey the three distributions for those higher than the significance level. According to these results, the experimental data of AC BDV for the 2 mm electrode gap fit better to the normal distribution than the EV and Weibull distributions. In addition, the EV and Weibull gave quite the same *p*-values since the Weibull is a particular case of EV [13]. 

Figure 11, Figure 12, Figure 13, Figure 14, Figure 15 and Figure 16 show the normal and EV probability density plot of Fe_3_O_4_ (20 nm), ZnO (20 nm), ZrO_2_ (20–30 nm), SiO_2_ (10–20 nm), Al_2_O_3_ (20–30 nm), and Al_2_O_3_ (50 nm) NFs versus AC BDV for different concentrations; they show how the breakdown data fit each case’s corresponding normal and EV probability lines. From those plots, the BDV at risk levels (1%, 10%, and 50%) are evaluated from normal and EV distribution fit curves and presented in Table 5a,b. The BDV at 1% and 10% risk levels (U_1%_ and U_10%_) are essential information about the reliability of the HV apparatuses since they represent their voltage limit for safe/continuous operation and the lowest possible AC BDV, while the BDV at 50% (U_50%_) is an estimation of the expected mean BDV [25]. 

It resorts from the results in Table 5a,b that the addition of the Fe_3_O_4_ (20 nm), ZnO (25 nm), ZrO_2_ (20–30 nm), SiO_2_ (10–20 nm), Al_2_O_3_ (20–30 nm), and Al_2_O_3_ (50 nm) NPs could not only enhance the mean AC BDV (from 50% risk level) but also improve the AC BDV at 1% and 10% risk levels. Mainly, the addition of these NPs strongly affects the U_1%_ rather than U_10%_ and U_50%_, exceeding 75% of improvement with Fe_3_O_4_ (20 nm) and Al_2_O_3_ (20–30 and 50 nm) compared to pure SE. Still, Fe_3_O_4_ (20 nm) and Al_2_O_3_ (20–30 and 50 nm) give the best U_10%_ and U_50%_ compared to ZnO (25 nm), ZrO_2_ (20–30 nm), and SiO_2_ (10–20 nm) NFs.

#### 3.4.2. Statistical Analysis of AC Breakdown Voltage Outcomes for 2.5 mm Gap Distance

Following the same steps presented in the previous subsection, the *p*-value for the EV, normal, and Weibull distributions were calculated and compared to the significance level for AC BDV data for 2.5 mm electrode gap (Table 6). The concerned NPs are ZnO (25 nm), ZrO_2_ (20–30 nm), and SiO_2_ (10–20 nm). It was noted that the *p*-value is higher than the significance level for all cases except ZrO_2_ (20–30 nm) NF at 0.1 g/L with normal distribution, and therefore, most of the experimental data obey the three distributions. 

Figure 17 and Figure 18 show the normal and EV probability density plot of ZnO (25 nm), ZrO_2_ (20–30 nm), and SiO_2_ (10–20 nm) NFs versus AC BDV for different concentrations. 

From those plots, the BDV at risk levels (1%, 10%, and 50%) are evaluated and presented in Table 7. Like those for 2 mm electrode gaps, those results for 2.5 mm show that the addition of ZnO (25 nm), ZrO_2_ (20–30 nm), and SiO_2_ (10–20 nm) NPs could not only enhance the mean AC BDV but also improve the AC BDV at 1% and 10% risk levels. Mainly, NPs’ addition affects the U_1%_ more than U_10%_ and U_50%_, exceeding 25% of improvement with the three NFs compared to pure SE. 

In addition, the concentrations for the optimal increments are identical to those that give the best mean AC BDV. This observation could easily be verified in Figure 17, Figure 18 and Figure 19 below, where the line corresponding to the optimal concentrations is in the right of the others, whatever the breakdown probability. 

### 3.5. Partial Discharge Activity of Al_2_O_3_ and Fe_3_O_4_ NFs 

Table 8 presents the average and standard deviation (St. Dev), as well as the increment percentage of PDIV (partial discharge inception voltage), PDEV (partial discharge extinction voltage), Q_avg_, Q_peak_, and NPDs/s values obtained from electrical measurements for different liquids tested with a threshold detection level of 500 fC. This threshold is just above the background noise and allows the PD activity comparison of the four liquids at the same voltage level. The higher applied voltage than 13 kV RMS could lead to the pure SE breakdown and for a threshold level higher than 500 fC, the PDIV and PDEV voltage could not be measured in the case of Al_2_O_3_ NFs. 

The PDIVs and PDEVs values of SE-based NFs are higher than those in pure SE in all cases. With 0.3 g/L Al_2_O_3_ (20–30 nm) NPs, the PDIV was enhanced by 24.58%, while 0.4 g/L Fe_3_O_4_ (20 nm) and Al_2_O_3_ (50 nm) NPs enhanced it by 22.95% and 24.14%, respectively. A similar tendency was observed with PDEV, i.e., 32.69%, 14.79%, and 28.66% of improvement in the case of Fe_3_O_4_ (20 nm), Al_2_O_3_ (20–30 nm), and Al_2_O_3_ (50 nm), respectively. Furthermore, a lower Q_avg_, Q_peak_, and NPDs/s for the three NFs than pure SE was observed. 

Figure 20, Figure 21, Figure 22 and Figure 23 show the PDs patterns of pure SE, SE-based Fe_3_O_4_ (20 nm), Al_2_O_3_ (20–30 nm), and Al_2_O_3_ (50 nm) NFs, respectively, at 13 kV (RMS) voltage level. It was observed that the PDs activity starts with the appearance of PDs at the peak of negative polarity (270° electrical degrees) and just a few cycles later at the peak of positive polarity (90° electrical degrees). Except for Fe_3_O_4_ (20 nm) NF, a smaller number of PDs was noticed in the positive polarity than in the negative, but with a higher charge level for pure SE, Al_2_O_3_ (20–30 nm), and Al_2_O_3_ (50 nm) NFs. The Fe_3_O_4_ (20 nm) NF manifests the lowest/highest activity in negative and positive polarities, respectively, unlike other NFs.

## 4. Discussion

Statistical analysis was performed on the AC BDV outcomes at 2 mm and 2.5 mm electrode gaps, and conformity to EV, normal, and Weibull statistical laws were conducted based on the *p*-value calculation. In other words, the AD statistics were employed to compute the corresponding *p*-value for each sample and compare it to the significance level. The results show that the experimental data of breakdown voltages for SE and SE-based Fe_3_O_4_ (20nm), ZnO (25 nm), ZrO_2_ (20–30 nm), SiO_2_ (10–20 nm), Al_2_O_3_ (20–30 nm), and Al_2_O_3_ (50 nm) NPs mostly obey EV, normal, and Weibull distributions. 

On the other hand, Fe_3_O_4_ (20 nm), ZnO (25 nm), ZrO_2_ (20–30 nm), SiO_2_ (10–20 nm), and Al_2_O_3_ (20–30 and 50 nm) NPs have a significant improvement of the AC breakdown voltage (AC BDV) of synthetic ester, MIDEL 7131. Furthermore, the BDV test was carried out for all NPs with a 2 mm electrode gap because of the breakdown voltage of Fe_3_O_4_ (20 nm) Al_2_O_3_ (20–30 and 50 nm) NFs for a 2.5 mm electrode gap exceeds the tester’s voltage limitation, which is 100 kV. For the 2 mm electrode gaps, SiO_2_ (10–20 nm) and Al_2_O_3_ (20–30 nm) NPs give the best improvements at a concentration of 0.3 g/L, while the best improvements with Fe_3_O_4_ (20 nm), ZnO (25 nm), ZrO_2_ (20–30 nm), and Al_2_O_3_ (50 nm) NPs are observed with a concentration 0.4 g/L. Similar results have been reported by other researchers [14,16,17,31]. 

The fact that there is an optimal concentration of NPs could be due to the saturation of the NPs/SE interfaces [16]; hence, the smaller particles may show their best AC BDV at lower concentrations than bigger particles. Beyond optimal concentration, an additional amount of NPs will show a lower or even negative effect on the AC BDV. Accordingly, up to 0.3 g/L, a slightly lower mean AC BDV has been observed with Al_2_O_3_ (50 nm) than with Al_2_O_3_ (20–30 nm); a reversed tendency is observed beyond 0.4 g/L. Khaled and Beroual reported a similar tendency with mineral oil and SE [16]. With Fe_3_O_4_ (20 nm), ZnO (25 nm), ZrO_2_ (20–30 nm), and Al_2_O_3_ (50 nm) NFs, the best enhancements are about 37.70%, 19.38%, 21.37%, and 42.13%, respectively, while there are about 20.30% and 44.12% in the case of SiO_2_ (10–20 nm) and Al_2_O_3_ (20–30 nm) NFs, respectively, compared to pure SE. From the results presented above, Fe_3_O_4_ (20 nm) and Al_2_O_3_ (20–30 nm and 50 nm) NFs show the best performances according to AC BDV results. The best improvements are greater than 35% for all cases. 

Nevertheless, a minor decline in AC BDV of ZnO (25 nm), ZrO_2_ (20–30 nm), and SiO_2_ (10–20 nm) NFs is observed after modifying the electrodes gap to 2.5 mm. Furthermore, the optimal concentrations of those NPs are confirmed, 0.4 g/L for ZnO (25 nm), ZrO_2_ (20–30 nm) and NPs, and 0.3 g/L for SiO_2_ (10–20 nm) NPs, which give 14.28%, 11.13%, and 12.83% of improvements compared to SE, respectively.

Khaled and Beroual [16] examined the same SE MIDEL 7131 with Fe_3_O_4_ (50 nm), SiO_2_ (10–20 nm), and Al_2_O_3_ (13 and 50 nm) NPs. They reported that the best improvements in breakdown voltages are obtained with Fe_3_O_4_ and SiO_2_ NFs at the maximum concentration of 0.4 g/L (upper limit, no optimal concentration), with Al_2_O_3_ (50 nm) NF at 0.3 g/L (optimal concentration), and, at 0.05 g/L (lower limit, minimum concentration) with Al_2_O_3_ (13 nm) NF. Their work suggests that the smaller the NPs, the higher AC BDV is (for the same concentration). The best improvement with SiO_2_ NF is about 30% compared to pure SE, while in the present work, the optimal improvement is about 11.62% at 0.3 g/L (for 2.5 mm electrode gap). In addition, they reported a lower AC BDV breakdown voltage for a MIDEL 7131 (60 kV) compared to the results in this work (82.6 kV); likely, this is likely because they have been used an aged MIDEL 7131. The same authors investigated the effect of conductive NPs (Fe_3_O_4_) on the AC BDV of mineral oil, synthetic and natural Esters-based NFs [34]. Their findings reveal that Fe_3_O_4_ nanoparticles significantly improve AC BDV of mineral oil (MO) and synthetic ester (SE). These improvements are 100% and 48% with MO and SE-based NFs, respectively. The improvement does not exceed 7% with natural ester, unlike MO and SE, even reducing the AC BDV. 

Different mechanisms have been proposed to describe the processes associated with a dielectric strength enhancement when adding a small amount of NPs to host liquids. Hwang et al. [11] introduced the electron scavenging model as a possible mechanism that depends on the relaxation time constant (τ_r_). It well describes the enhancement of the insulating performances of base liquid with addressed conductive NPs (Fe_3_O_4_), but it fails to explain the performance improvement of other NPs, even conductive ones [18]. The mechanisms by which conductive, semi-conductive, and dielectric NPs trap electrons are explained by the potential well distribution caused by induced or polarized charges [10]. The difference between conductivity or permittivity of NPs and host liquid could generate induced and/or polarized charges on the NPs/SE interface, producing electrons trapping site [10]. The formed trapping site on the interface could trap moving electrons, enhancing the base liquid’s breakdown performance.

The involved mechanisms in enhancing BDV of NFs could also be discussed by considering the behavior of spherical particles (roughly assuming that the particles are spherical) suspended in a liquid and subjected to a uniform applied electric field; three cases are possible.

The first case is when the polarizability of the NP is more significant than the base liquid. That means that there are more charges inside the interface (NP side) than outside (liquid side), resulting in a surface charge density difference on both sides of the interface. So, an induced dipole is aligned with the field applied through the particle [35]. A suitable example of this case would be a conducting NP (or NP ‘insulting or conducting’ with a high dielectric constant) in an insulating liquid with a low dielectric constant. 

The second case is when the polarizability of the NP is less significant than the base liquid, which means that there are fewer charges inside the interface (NP side) than outside (liquid side). The resulted dipole points in the opposite direction. This could be an insulating NP suspended in a liquid with a high dielectric constant or high conductivity. 

The third case is when the polarizability of the NP and that of the liquid are the same, and there is no net dipole. Obviously, the second and the third cases do not correspond to our study since, in most cases, the NPs conductivity and/or permittivity is higher than that of SE. 

Nevertheless, the charge polarized on the surface of the NP produces the trapping site (potential well) [10]. Electron trapping by NPs significantly slows down the streamer’s development by reducing its velocity and enhancing the NFs breakdown voltage. Sima et al. [10] conducted studies investigating the depth of the potential well of conducting Fe_3_O_4_, semi-conductive TiO_2_, and insulating Al_2_O_3_ NPs. They found that the potential wells of dielectric NPs are shallower than those of conductive NPs [10]. 

The results from the performed PDs test on Fe_3_O_4_ (20 nm), Al_2_O_3_ (20–30 nm), and Al_2_O_3_ (50 nm) nanofluids at optimal concentrations show a considerable enhancement of PDs resistivity of pure SE. Alike, it was found that the PD inception voltage (PDIV), PD extinction voltage (PDEV) are pushed to higher voltage levels when adding Fe_3_O_4_ (20 nm), Al_2_O_3_ (20–30 nm), and Al_2_O_3_ (50 nm) NPs to the reference SE under AC stress. Also, those NFs show a reduced Q_avg_, Q_peak_, and NPDs/s compared to SE. 

According to Atiya et al. [24], the thickness of the electrical double layer (EDL) around NP plays a vital role in reducing PD activity; the thicker the EDL, the more resistive to PD. However, in our case, the zeta potential measurements show that Al_2_O_3_ (50 nm) NPs show a higher zeta potential than Fe_3_O_4_ (20 nm) and Al_2_O_3_ (20–30 nm) NPs; hence the EDL thickness for Al_2_O_3_ (50 nm) NPs is larger than that for Al_2_O_3_ (20–30 nm) and Fe_3_O_4_ (20 nm) NPs [24].

Depending on the properties of the NPs and the liquid used, the presence of NPs can induce electric field heterogeneity in the NF. The localized increase of the electric field can lead to PDs at the same applied voltage or at the same ionization level of the liquid. Thus, near the electrodes, high mobility electrons and low mobility ionized ions are ready to migrate under the electric field forces. The NPs then play the role of electrons and negative ion scavengers, which generate a potential well (trapping site) that reduces the electron/negative ion movement and thence PD activities [10,36]. When the trapping process is finished, nanoparticle surfaces are saturated with negative charges; hence they no longer could trap more electrons. The limit is strongly correlated with the mismatch between base liquid and NP’s conductivities and/or permittivities [10,35].

## 5. Conclusions

In this work, it was shown that the AC breakdown voltage of synthetic ester-based ZrO_2_ (20–30 nm), ZnO (25 nm), SiO_2_ (10–20 nm), Al_2_O_3_ (20–30 nm), Al_2_O_3_ (50 nm), and Fe_3_O_4_ (20 nm) nanofluids are improved. The improvements are the best with conductive nanoparticles (Fe_3_O_4_) and insulating nanoparticles (Al_2_O_3_), regardless of the size. However, the addition of insulating nanoparticles (Al_2_O_3_) with the smallest size (20–30 nm) shows the best improvement at a lower concentration. The statistical analysis of the experimental results shows that the breakdown voltage outcomes mostly obey the EV, normal, and Weibull distributions. Additionally, it has been shown that the addition of Al_2_O_3_ (20–30 nm), Al_2_O_3_ (50 nm), and Fe_3_O_4_ (20 nm) NPs significantly reduces the partial discharge activity compared to pure SE.

## Figures and Tables

**Figure 1 nanomaterials-12-02105-f001:**
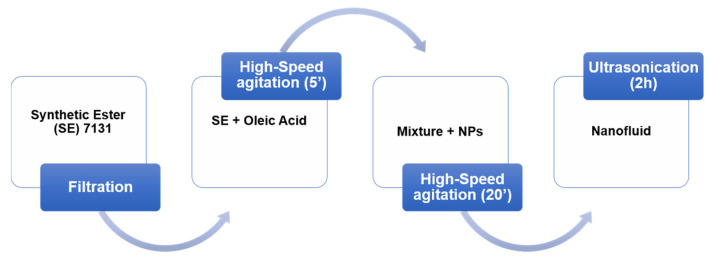
The preparation procedure of nanofluids (NFs).

**Figure 2 nanomaterials-12-02105-f002:**
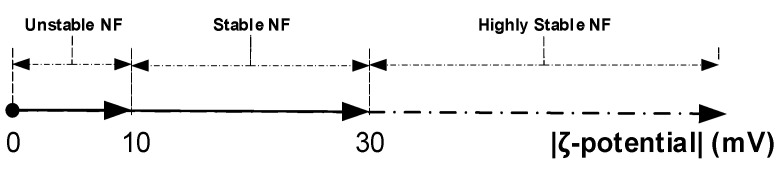
Significance of zeta potential values.

**Figure 3 nanomaterials-12-02105-f003:**
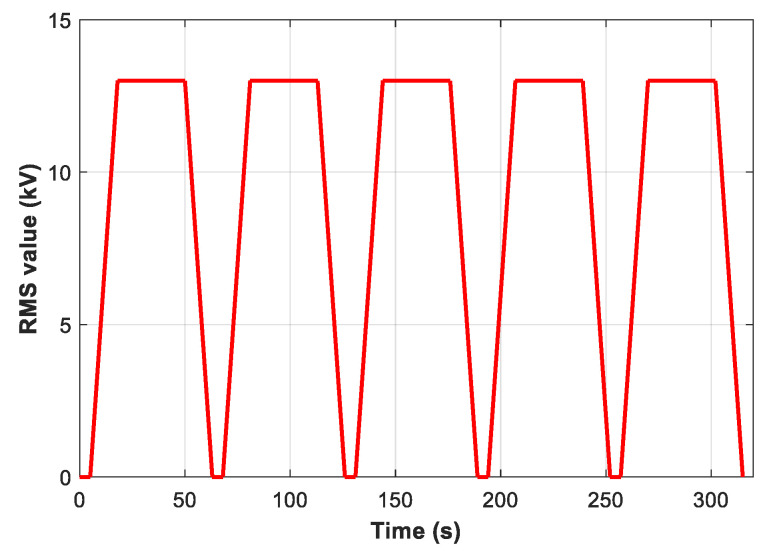
Voltage profile (RMS value) applied to samples during partial discharge test.

**Figure 4 nanomaterials-12-02105-f004:**
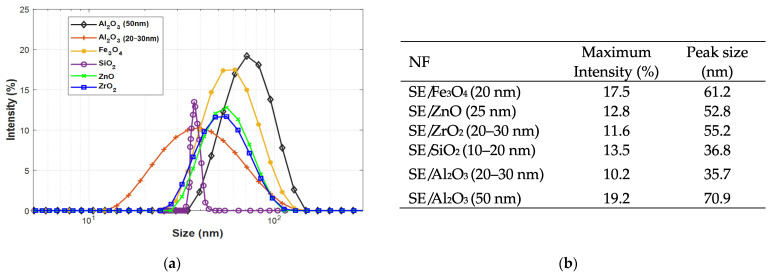
Nanoparticle size distribution (**a**,**b**).

**Figure 5 nanomaterials-12-02105-f005:**
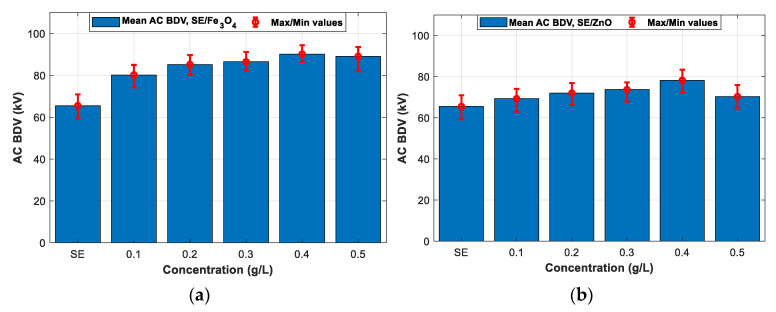
Mean breakdown voltages of synthetic ester and synthetic ester-based nanofluid at different concentrations of nanoparticles: (**a**) Fe_3_O_4_ (20 nm), (**b**) ZnO (25 nm), (**c**) ZrO_2_ (20–30 nm), (**d**) SiO_2_ (10–20 nm), (**e**) Al_2_O_3_ (20–30 nm), (**f**) Al_2_O_3_ (50 nm).

**Figure 6 nanomaterials-12-02105-f006:**
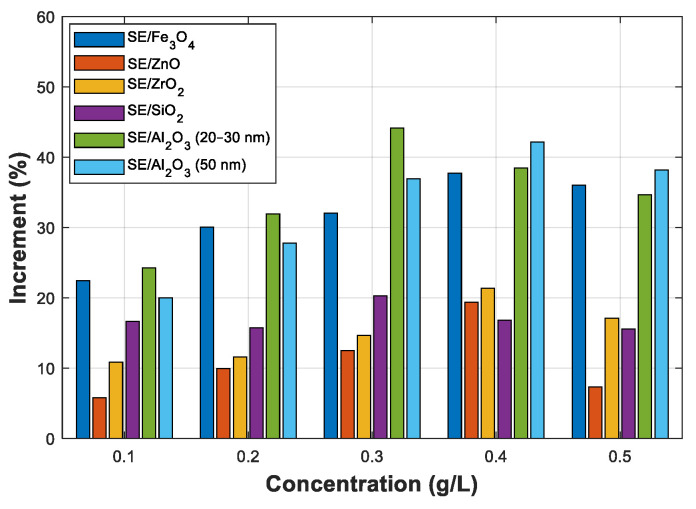
Comparison between the improvements of breakdown voltages of different nanofluids for different concentrations.

**Figure 7 nanomaterials-12-02105-f007:**
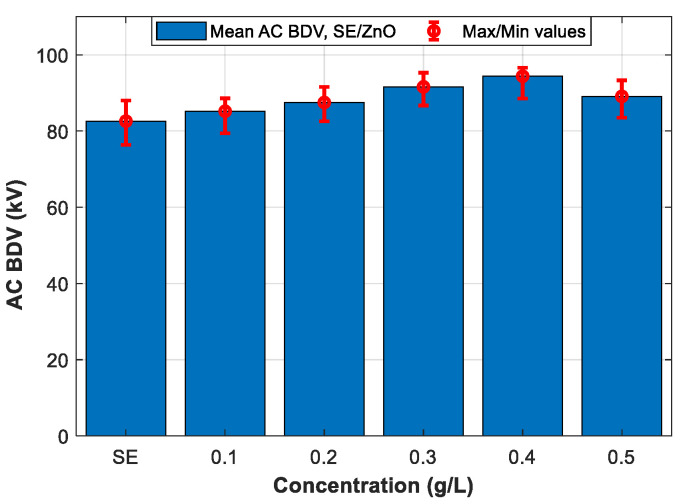
Mean breakdown voltages of synthetic ester and synthetic ester-based ZnO (25 nm) nanofluids at different concentrations.

**Figure 8 nanomaterials-12-02105-f008:**
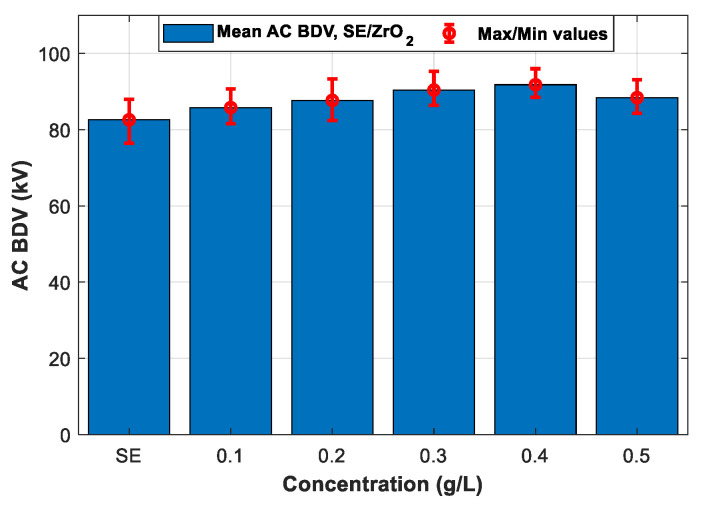
Mean breakdown voltages of synthetic ester and synthetic ester-based ZrO_2_ (20–30 nm) nanofluids at different concentrations.

**Figure 9 nanomaterials-12-02105-f009:**
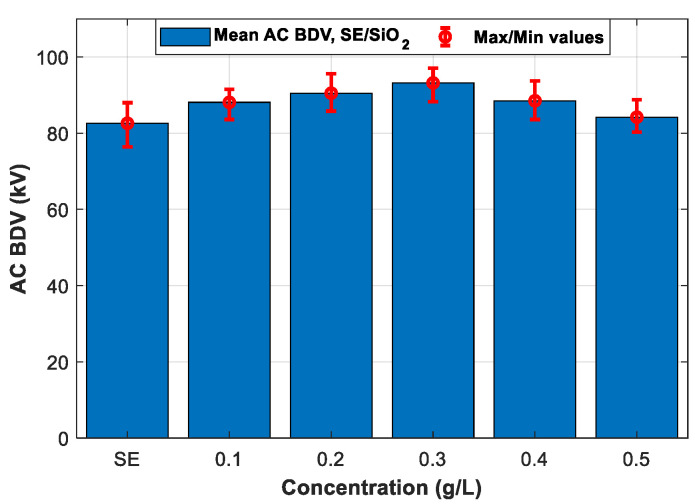
Mean breakdown voltages of synthetic ester and synthetic ester-based SiO_2_ (10–20 nm) nanofluids at different concentrations.

**Figure 10 nanomaterials-12-02105-f010:**
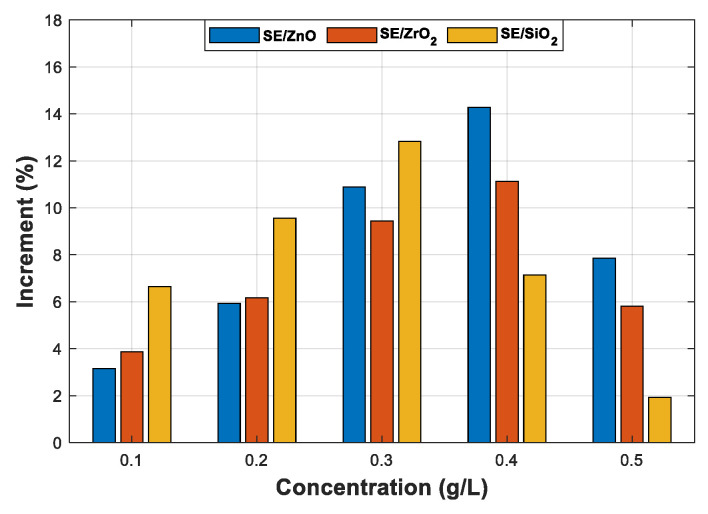
Comparison between the improvements of breakdown voltages of different nanofluids for different concentrations.

**Figure 11 nanomaterials-12-02105-f011:**
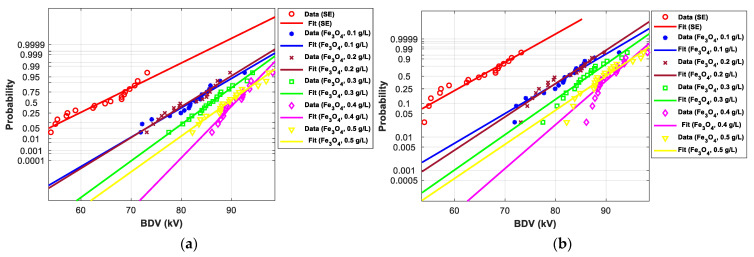
Probability plot of breakdown voltage data of SE-based nanofluids with Fe_3_O_4_ (20 nm) for the 2 mm electrode gap; (**a**) normal and (**b**) extreme value.

**Figure 12 nanomaterials-12-02105-f012:**
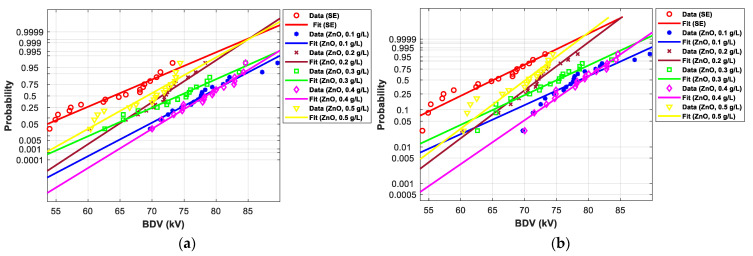
Probability plot of breakdown voltage data of SE-based nanofluids with ZnO (25 nm) for 2 mm electrode gap; (**a**) normal and (**b**) extreme value.

**Figure 13 nanomaterials-12-02105-f013:**
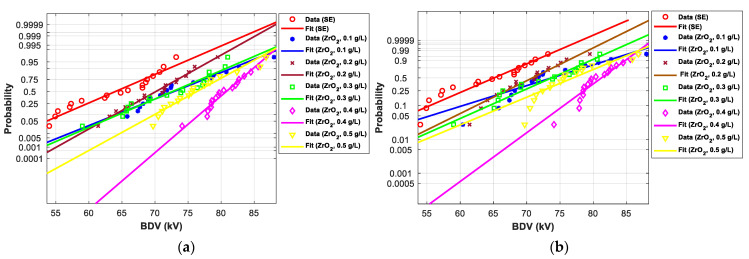
Probability plot of breakdown voltage data of SE-based nanofluids with ZrO_2_ (20–30 nm) for the 2 mm electrode gap; (**a**) normal and (**b**) extreme value.

**Figure 14 nanomaterials-12-02105-f014:**
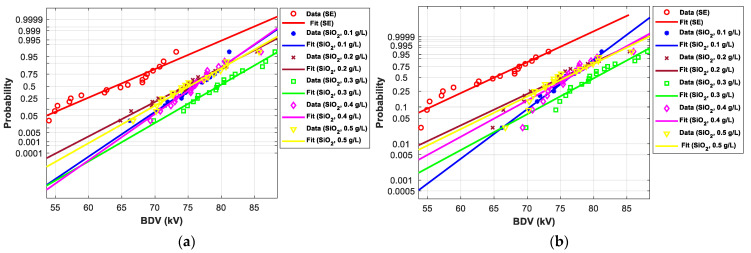
Probability plot of breakdown voltage data of SE-based nanofluids with SiO_2_ (10–20 nm) for the 2 mm electrode gap; (**a**) normal and (**b**) extreme value.

**Figure 15 nanomaterials-12-02105-f015:**
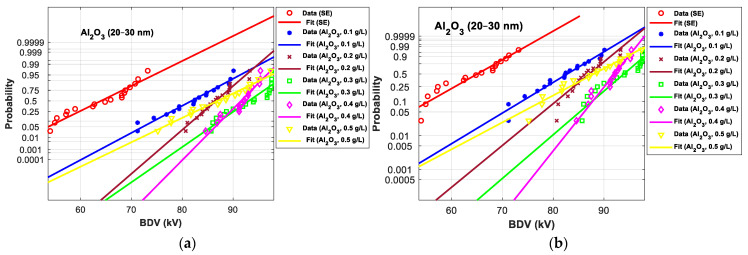
Probability plot of breakdown voltage data of SE-based nanofluids with Al_2_O_3_ (20–30 nm) for the 2 mm electrode gaps; (**a**) normal and (**b**) extreme value.

**Figure 16 nanomaterials-12-02105-f016:**
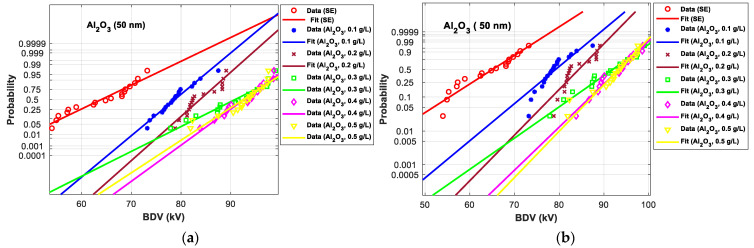
Probability plot of breakdown voltage data of SE-based nanofluids with Al_2_O_3_ (50 nm) for the 2 mm electrode gap; (**a**) normal and (**b**) extreme value.

**Figure 17 nanomaterials-12-02105-f017:**
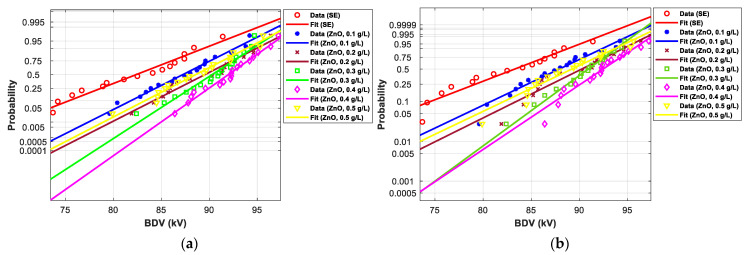
Probability plot of breakdown voltage data of SE-based nanofluids with ZnO (25 nm) for 2.5 mm electrode gaps; (**a**) normal and (**b**) extreme value.

**Figure 18 nanomaterials-12-02105-f018:**
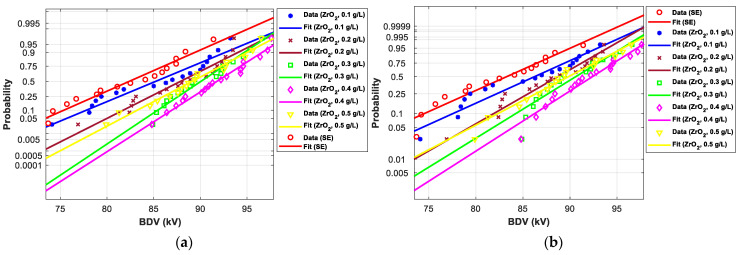
Probability plot of breakdown voltage data of SE-based nanofluids with ZrO_2_ (20–30 nm) for 2.5 mm electrode gaps; (**a**) normal and (**b**) extreme value.

**Figure 19 nanomaterials-12-02105-f019:**
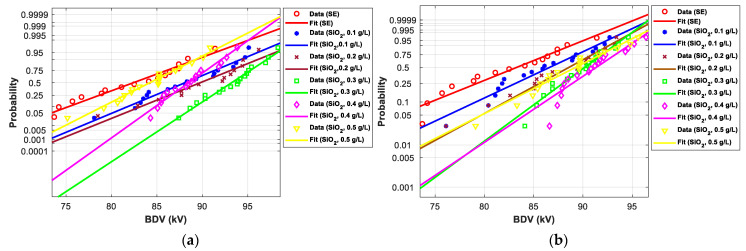
Probability plot of breakdown voltage data of SE-based nanofluids with SiO_2_ (10–20 nm) for 2.5 mm electrode gaps; (**a**) normal and (**b**) extreme value.

**Figure 20 nanomaterials-12-02105-f020:**
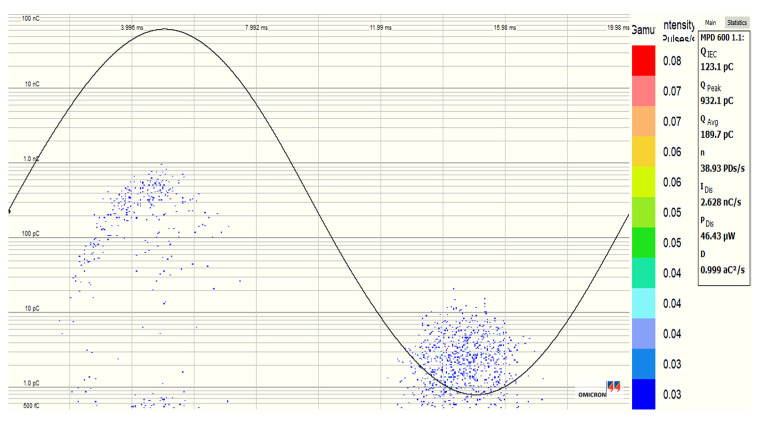
PD pattern of pure synthetic ester (MIDEL 7131) at 13 kV (RMS) voltage level.

**Figure 21 nanomaterials-12-02105-f021:**
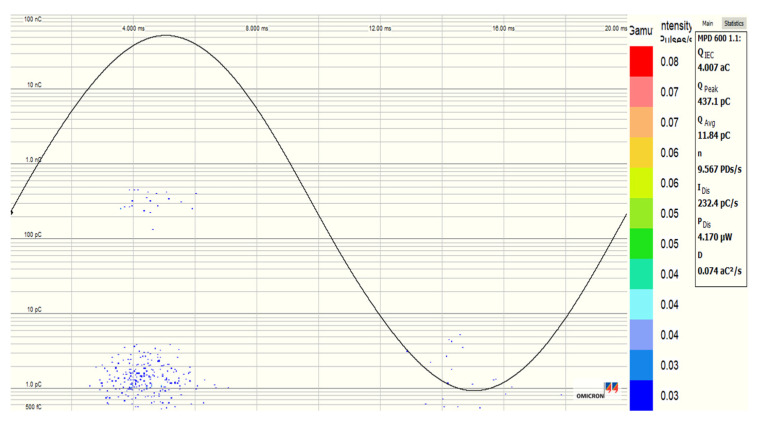
PD pattern of synthetic ester-based NF with Fe_3_O_4_ (20 nm, 0.4 g/L) at 13 kV (RMS) voltage level.

**Figure 22 nanomaterials-12-02105-f022:**
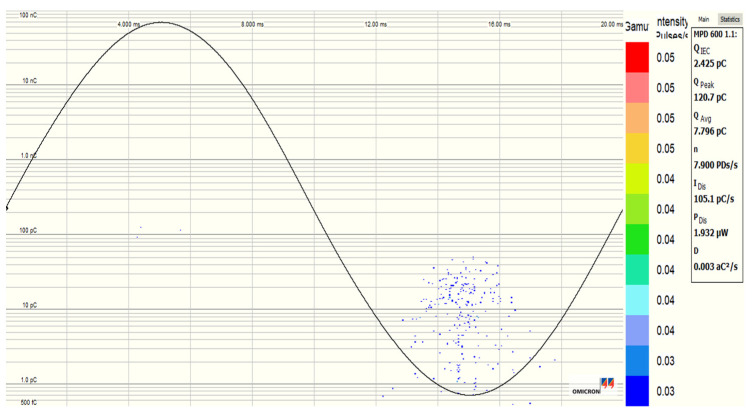
PD pattern of synthetic ester-based NF with Al_2_O_3_ (20–30 nm, 0.3 g/L) at 13 kV (RMS) voltage level.

**Figure 23 nanomaterials-12-02105-f023:**
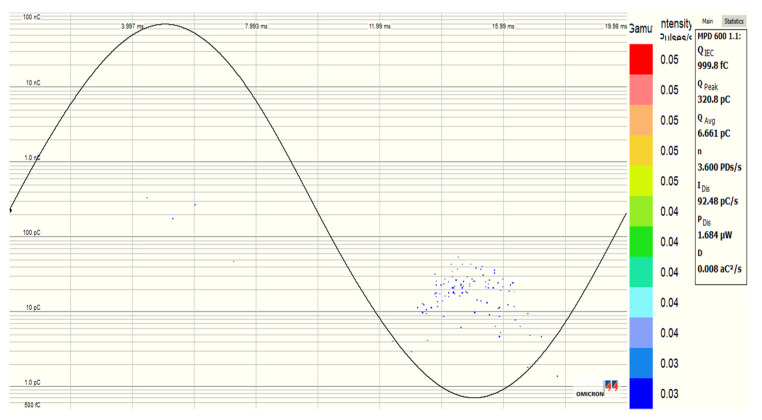
PD pattern of synthetic ester-based NF with Al_2_O_3_ (50 nm, 0.4 g/L) at 13 kV (RMS) voltage level.

**Table 1 nanomaterials-12-02105-t001:** Physicochemical Properties of Synthetic Ester, MIDEL 7131.

Property	MIDEL 7131
Density at 20 °C (kg/L)	0.97
Kinematic Viscosity (mm^2^/s)	
at 40 °C	29
at −20 °C	1440
Pour point (°C)	−56
Flash point (°C)	260
Fire point (°C)	316
Water content (ppm)	50
AC BDV “60Hz” (kV)	>75
Dielectric constant	3.2
Power Factor at 90 °C	<0.008
DC Resistivity at 90 °C (GΩ·m)	>20

**Table 2 nanomaterials-12-02105-t002:** Characteristics of the nanoparticles.

NPs	Fe_3_O_4_	ZnO	ZrO_2_	SiO_2_	Al_2_O_3_	Al_2_O_3_
Size (nm)	20	25	20–30	10–20	20–30	50
Specific surface area (m^2^/g)	40	19 ± 5	15–35	60–100	120–140	80
Density (g/cm^3^)	5.1 *	5.6	4.8–6.0	0.6~1.8	3–3.98 *	
Dielectric Constant	80 *	8.5	10–23	3.9	9–10.1 *	
Purity (%)	99.5	99.5	99	99.8	99.9	99.9

* The values are taken from literature as they are not provided by the suppliers.

**Table 3 nanomaterials-12-02105-t003:** Zeta potential and electrical conductivity of nanofluids samples.

	NFs	Zeta Potential (mV)	Elec. Conductivity (mS/cm) × 10^−3^
(0.1 g/L)	SE/Fe_3_O_4_ (20 nm)	−22.4	2.16
	SE/ZnO (25 nm)	+22.9	0.94
	SE/ZrO_2_ (20–30 nm)	−24.4	2.09
	SE/SiO_2_ (10–20 nm)	−36.2	1.73
	SE/Al_2_O_3_ (20–30 nm)	+15.9	1.24
	SE/Al_2_O_3_ (50 nm)	+24.9	1.37
(0.2 g/L)	SE/Fe_3_O_4_ (20 nm)	−23.6	2.31
	SE/ZnO (25 nm)	+19.7	1.26
	SE/ZrO_2_ (20–30 nm)	−21.8	1.89
	SE/SiO_2_ (10–20 nm)	−33.7	1.57
	SE/Al_2_O_3_ (20–30nm)	+20.4	0.85
	SE/Al_2_O_3_ (50 nm)	+29.0	1.06
(0.3 g/L)	SE/Fe_3_O_4_ (20 nm)	−17.9	2.78
	SE/ZnO (25 nm)	+17.0	1.41
	SE/ZrO_2_ (20–30 nm)	-	-
	SE/SiO_2_ (10–20 nm)	−31.9	1.02
	SE/Al_2_O_3_ (20–30 nm)	-	-
	SE/Al_2_O_3_ (50 nm)	+21.0	0.67

**Table 4 nanomaterials-12-02105-t004:** Hypothesis test of conformity of breakdown voltage outcomes of various nanofluids to EV, normal, and Weibull distributions considering *p*-value calculation for the 2 mm electrode gaps.

	EV		Normal		Weibull	
Concentration	*p*-Value	Decision	*p*-Value	Decision	*p*-Value	Decision
Pure SE	0.2621	Accepted	0.1175	Accepted	0.2020	Accepted
SE-based NFs with Fe_3_O_4_ (20 nm)						
(0.1 g/L)	0.3170	Accepted	0.2484	Accepted	0.4167	Accepted
(0.2 g/L)	0.2473	Accepted	0.2256	Accepted	0.2436	Accepted
(0.3 g/L)	0.2993	Accepted	0.9808	Accepted	0.4374	Accepted
(0.4 g/L)	0.0262	Not-Accepted	0.2699	Accepted	0.0387	Not-Accepted
(0.5 g/L)	0.0986	Accepted	0.5949	Accepted	0.1481	Accepted
SE-based NFs with ZnO (25 nm)						
(0.1 g/L)	0.0250	Not-Accepted	0.5068	Accepted	0.0564	Accepted
(0.2 g/L)	0.3274	Accepted	0.1113	Accepted	0.3743	Accepted
(0.3 g/L)	0.5768	Accepted	0.6064	Accepted	0.7095	Accepted
(0.4 g/L)	0.5845	Accepted	0.5523	Accepted	0.6087	Accepted
(0.5 g/L)	0.0410	Not-Accepted	0.0050	Not-Accepted	0.0247	Not-Accepted
SE-based NFs with ZrO_2_ (20–30 nm)						
(0.1 g/L)	0.0394	Not-Accepted	0.3584	Accepted	0.0889	Accepted
(0.2 g/L)	0.7796	Accepted	0.9230	Accepted	0.8879	Accepted
(0.3 g/L)	0.2449	Accepted	0.1495	Accepted	0.2099	Accepted
(0.4 g/L)	0.1931	Accepted	0.5805	Accepted	0.2398	Accepted
(0.5 g/L)	0.0198	Not-Accepted	0.3314	Accepted	0.0408	Not-Accepted
SE-based NFs with SiO_2_ (25 nm)						
(0.1 g/L)	0.9152	Accepted	0.7484	Accepted	0.9302	Accepted
(0.2 g/L)	0.3551	Accepted	0.8744	Accepted	0.5429	Accepted
(0.3 g/L)	0.2302	Accepted	0.8775	Accepted	0.3447	Accepted
(0.4 g/L)	0.0170	Not-Accepted	0.5653	Accepted	0.0381	Not-Accepted
(0.5 g/L)	0.1700	Accepted	0.7144	Accepted	0.2678	Accepted
SE-based NFs with Al_2_O_3_ (20–30 nm)						
(0.1 g/L)	0.8544	Accepted	0.6733	Accepted	0.8801	Accepted
(0.2 g/L)	0.1537	Accepted	0.8117	Accepted	0.2183	Accepted
(0.3 g/L)	0.0068	Not-Accepted	0.0111	Not-Accepted	0.0066	Not-Accepted
(0.4 g/L)	0.3330	Accepted	0.0459	Not-Accepted	0.3060	Accepted
(0.5 g/L)	0.8477	Accepted	0.6342	Accepted	0.8440	Accepted
SE-based NFs with Al_2_O_3_ (50 nm)						
(0.1 g/L)	0.0363	Accepted	0.6200	Accepted	0.0644	Accepted
(0.2 g/L)	0.0009	Not-Accepted	0.0056	Not-Accepted	0.0012	Not-Accepted
(0.3 g/L)	0.7286	Accepted	0.3314	Accepted	0.6836	Accepted
(0.4 g/L)	0.5240	Accepted	0.2928	Accepted	0.5125	Accepted
(0.5 g/L)	0.4755	Accepted	0.0203	Not-Accepted	0.3848	Accepted

**Table 5 nanomaterials-12-02105-t005:** The AC BDV at 1, 10, and 50% probability levels evaluated from normal and extreme value fit curves.

		1%		10%		50%	
Concentration	Concentration	BDV (kV)	Increment (%)	BDV (kV)	Increment (%)	BDV (kV)	Increment (%)
Pure SE	Normal	49.30	-	55.90	-	64.00	-
	EV	43.60	-	55.60	-	65.10	-
SE-based NFs with Fe_3_O_4_ (20 nm)							
(0.1 g/L)	Normal	69.50	40.97	75.10	34.34	81.90	27.96
	EV	62.10	42.43	73.50	32.19	82.60	26.88
(0.2 g/L)	Normal	69.70	41.37	74.90	33.98	81.40	21.18
	EV	64.00	46.79	74.10	33.27	82.30	26.42
(0.3 g/L)	Normal	76.10	54.36	80.60	44.18	86.00	34.37
	EV	69.30	58.94	78.90	41.91	86.60	33.03
(0.4 g/L)	Normal	83.70	69.77	86.90	55.45	90.80	41.87
	EV	77.20	77.06	84.90	52.70	91.10	39.94
(0.5 g/L)	Normal	79.30	60.85	83.80	49.91	89.40	39.78
	EV	72.20	65.60	82.10	47.66	90.00	38.25
SE-based NFs with ZnO (25 nm)							
(0.1 g/L)	Normal	66.10	34.07	71.50	27.90	78.00	21.87
	EV	55.60	27.52	68.40	23.02	78.70	20.89
(0.2 g/L)	Normal	62.00	25.76	66.30	18.60	71.60	11.87
	EV	58.20	33.49	66.00	18.71	72.20	10.91
(0.3 g/L)	Normal	60.20	22.10	66.50	18.96	74.20	15.93
	EV	52.60	20.64	65.10	17.09	75.10	15.36
(0.4 g/L)	Normal	68.20	38.33	72.80	30.23	78.40	22.5
	EV	63.90	46.56	72.40	30.22	79.20	21.66
(0.5 g/L)	Normal	58.10	17.84	63.10	12.88	69.30	8.28
	EV	56.20	28.90	64.00	15.11	70.20	7.83
SE-based NFs with ZrO_2_ (20–30 nm)							
(0.1 g/L)	Normal	57.60	16.83	64.20	14.84	72.90	13.90
	EV	44.00	0.92	60.50	8.81	73.80	13.36
(0.2 g/L)	Normal	58.80	19.26	63.90	14.31	70.20	9.68
	EV	52.10	19.50	62.50	12.41	70.90	8.91
(0.3 g/L)	Normal	57.90	17.44	64.50	15.38	72.70	13.59
	EV	52.90	21.33	64.50	16.01	73.80	13.36
(0.4 g/L)	Normal	73.40	48.88	76.80	37.38	80.80	26.25
	EV	68.60	57.34	75.70	36.15	81.30	24.88
(0.5 g/L)	Normal	64.80	31.44	70.00	25.22	76.40	19.37
	EV	54.70	25.46	67.10	20.68	77.00	18.28
SE-based NFs with SiO_2_ (25 nm)							
(0.1 g/L)	Normal	66.40	34.68	70.60	26.29	75.80	18.43
	EV	63.00	44.50	70.50	26.80	76.40	17.36
(0.2 g/L)	Normal	62.50	26.77	67.90	21.46	74.60	16.56
	EV	54.10	24.08	65.90	18.53	75.30	15.67
(0.3 g/L)	Normal	68.70	39.35	73.60	31.66	79.60	24.35
	EV	61.80	41.74	72.10	29.68	80.30	23.35
(0.4 g/L)	Normal	67.00	35.90	71.00	27.01	76.00	18.75
	EV	58.10	33.26	68.30	22.84	76.40	17.36
(0.5 g/L)	Normal	64.10	30.02	69.10	23.61	75.30	17.65
	EV	55.50	27.29	66.80	20.14	75.90	16.59
SE-based NFs with Al_2_O_3_ (20–30 nm)							
(0.1 g/L)	Normal	68.30	38.53	74.40	33.09	81.90	27.96
	EV	62.60	43.58	73.80	32.73	82.80	27.19
(0.2 g/L)	Normal	68.30	38.53	82.00	46.69	86.70	35.46
	EV	72.40	66.06	80.60	44.96	87.20	33.95
(0.3 g/L)	Normal	82.70	67.74	87.40	56.35	93.20	45.62
	EV	79.80	83.03	87.70	57.73	94.00	44.39
(0.4 g/L)	Normal	84.50	71.39	87.70	56.88	91.60	43.125
	EV	82.40	88.99	87.80	57.91	92.10	41.47
(0.5 g/L)	Normal	72.00	46.04	78.70	40.78	87.00	35.93
	EV	65.10	49.31	77.80	39.93	88.10	35.33
SE-based NFs with Al_2_O_3_ (50 nm)							
(0.1 g/L)	Normal	70.30	42.59	74.20	32.73	78.80	23.12
	EV	62.70	43.81	71.90	29.32	79.20	21.66
(0.2 g/L)	Normal	76.00	54.15	79.50	42.21	83.90	31.09
	EV	70.90	62.61	78.40	41.01	84.40	29.65
(0.3 g/L)	Normal	76.90	55.98	83.20	48.83	91.00	42.18
	EV	72.30	65.83	83.30	49.82	92.00	41.32
(0.4 g/L)	Normal	83.00	68.35	87.40	56.35	92.70	44.84
	EV	79.10	81.42	87.10	56.65	93.40	43.47
(0.5 g/L)	Normal	81.80	65.92	86.70	55.09	92.60	44.68
	EV	80.10	83.72	87.50	57.37	93.40	43.47

**Table 6 nanomaterials-12-02105-t006:** Hypothesis test of conformity of breakdown voltage outcomes of various nanofluids to EV, Normal, and Weibull distributions considering *p*-value calculation, for 2.5 mm electrode gaps.

	EV		Normal		Weibull	
Concentration	*p*-Value	Decision	*p*-Value	Decision	*p*-Value	Decision
Pure SE	0.5275	Accepted	0.5750	Accepted	0.5733	Accepted
SE-based NFs with ZnO (25 nm)						
(0.1 g/L)	0.7823	Accepted	0.9705	Accepted	0.8702	Accepted
(0.2 g/L)	0.3589	Accepted	0.2998	Accepted	0.3613	Accepted
(0.3 g/L)	0.9900	Accepted	0.4293	Accepted	0.9907	Accepted
(0.4 g/L)	0.6970	Accepted	0.6941	Accepted	0.7461	Accepted
(0.5 g/L)	0.4051	Accepted	0.8049	Accepted	0.4919	Accepted
SE-based NFs with ZrO_2_ (20–30 nm)						
(0.1 g/L)	0.0565	Accepted	0.0438	Not-Accepted	0.0507	Accepted
(0.2 g/L)	0.5412	Accepted	0.3017	Accepted	0.5162	Accepted
(0.3 g/L)	0.1277	Accepted	0.5189	Accepted	0.1612	Accepted
(0.4 g/L)	0.3543	Accepted	0.6686	Accepted	0.4006	Accepted
(0.5 g/L)	0.5753	Accepted	0.8512	Accepted	0.6822	Accepted
SE-based NFs with SiO_2_ (10–20 nm)						
(0.1 g/L)	0.4659	Accepted	0.6010	Accepted	0.5045	Accepted
(0.2 g/L)	0.7526	Accepted	0.3288	Accepted	0.7011	Accepted
(0.3 g/L)	0.8697	Accepted	0.7511	Accepted	0.8937	Accepted
(0.4 g/L)	0.1784	Accepted	0.5097	Accepted	0.2118	Accepted
(0.5 g/L)	0.4773	Accepted	0.7549	Accepted	0.5634	Accepted

**Table 7 nanomaterials-12-02105-t007:** The AC BDV at 1, 10, and 50% probability levels evaluated from normal and extreme value fit curves.

		1%		10%		50%	
Concentration	Distribution	BDV (kV)	Increment (%)	BDV (kV)	Increment (%)	BDV (kV)	Increment (%)
Pure SE	Normal	69.70	-	75.40	-	82.40	-
	EV	63.70	-	74.60	-	83.30	-
SE-based NFs with ZnO (25 nm)							
(0.1 g/L)	Normal	77.40	11.04	81.90	8.62	87.30	5.94
	EV	72.10	13.18	80.90	8.440	88.00	5.642
(0.2 g/L)	Normal	79.90	14.63	84.20	11.67	89.60	8.73
	EV	75.10	17.89	83.50	11.93	90.30	8.403
(0.3 g/L)	Normal	82.70	18.65	86.10	14.19	90.30	9.58
	EV	80.70	26.68	86.40	15.81	90.90	9.123
(0.4 g/L)	Normal	85.00	21.95	88.20	16.67	92.00	11.65
	EV	81.30	27.62	87.50	17.29	92.50	11.04
(0.5 g/L)	Normal	79.00	13.34	83.30	10.47	88.60	7.52
	EV	73.40	15.22	82.20	10.18	89.30	7.202
SE-based NFs with ZrO_2_ (20–30 nm)							
(0.1 g/L)	Normal	71.80	3.01	77.80	3.18	85.30	3.51
	EV	66.90	5.02	77.70	4.150	86.30	3.601
(0.2 g/L)	Normal	76.80	10.18	81.80	8.48	87.90	6.67
	EV	73.50	15.38	82.00	9.910	88.80	6.602
(0.3 g/L)	Normal	81.70	17.21	85.50	13.39	90.10	9.34
	EV	76.40	19.93	84.30	13.00	90.70	8.883
(0.4 g/L)	Normal	83.00	19.08	87.00	15.38	91.80	11.40
	EV	78.70	23.54	86.30	15.68	92.40	10.92
(0.5 g/L)	Normal	78.50	12.62	83.30	10.47	89.20	8.25
	EV	73.10	14.75	81.60	9.400	89.90	7.923
SE-based NFs with SiO_2_ (25 nm)							
(0.1 g/L)	Normal	76.80	10.18	81.90	8.62	88.10	6.91
	EV	71.60	12.40	81.20	8.84	88.90	6.722
(0.2 g/L)	Normal	78.10	12.05	83.40	10.61	89.80	8.98
	EV	75.10	17.89	83.70	12.19	90.70	8.883
(0.3 g/L)	Normal	86.10	23.52	89.30	18.43	93.30	13.22
	EV	82.60	29.67	88.80	19.03	93.70	12.48
(0.4 g/L)	Normal	82.00	17.64	84.90	12.59	88.50	7.40
	EV	77.50	21.66	83.90	12.46	88.90	6.722
(0.5 g/L)	Normal	75.00	7.60	79.10	4.90	84.20	2.18
	EV	70.10	10.04	78.30	4.95	84.90	1.920

**Table 8 nanomaterials-12-02105-t008:** PD activity of synthetic ester and synthetic ester-based nanofluids with Fe_3_O_4_ (20 nm) and Al_2_O_3_ (20–30 and 50 nm).

	Pure SE MIDEL 7131	Fe_3_O_4_ (20 nm) NF at 0.4 g/L	Al_2_O_3_ (20–30 nm) NF at 0.3 g/L	Al_2_O_3_ (50 nm) NF at 0.4 g/L
PDIV (kV)	10.286	12.647	12.815	12.77
St. Dev (kV)	0.763	1.0435	0.511	0.325
Increment (%)	-	22.95	24.58	24.14
PDEV (kV)	8.9226	11.840	10.243	11.48
St. Dev (kV)	2.1567	0.9922	1.352	1.054
Increment (%)	-	32.69	14.79	28.66
Qavr (pC)	123.8	3.30	3.933	4.15
St. Dev (pC)	20.06	1.574	0.377	5.864
Increment (%)	-	−97.33	−96.82	−96.64
Q_peak_ (pC)	740.44	122.68	48.95	510.6
St. Dev (pC)	49.250	97.07	29.23	104.20
Increment (%)	-	−83.43	−93.38	−31.04
NPDs/s (PDs/s)	9.20	1.00	1.16	1.16
St. Dev (PDs/s)	27.686	0.00	0.408	0.408
Increment (%)	-	−93.92	−90.14	−86.52

## Data Availability

The data reported in the present study are available upon request; please refer to the corresponding author when asking.

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
