# Peer review of "Effect of Conducting, Semi-Conducting and Insulating Nanoparticles on AC Breakdown Voltage and Partial Discharge Activity of Synthetic Ester: A Statistical Analysis"

_nanomaterials, 2022, doi:10.3390/nano12122105_

Round 1
Reviewer 1 Report
Work presentation is a little confusing. The introduction comments: “… influence of conducting (Fe3O4), semi-conductive (ZnO), and insulating (ZrO2, SiO2, and Al2O3) nanoparticles …”
But the order in which the results are presented does not follow this order of NPs; moreover, in different sections it follows different orders.
It would be appreciated if the results of the different tests of the NPs were always presented following the same order of NPs.
In some parts of the work (for example: Fig 4) the NPs sizes are included next to the name of the NPs, while in others the sizes are not included; The presentation of the work should be uniform.
Other aspect is that the work has an excessive number of tables and figures, those that provide redundant information should be eliminated, for example, Table 4 does not provide new information to that is presented in Figures 5 and 6; it is recommended to delete the table, or delete both figures.
Table 1: “Density at 20 (°C)”, density units are missed.
Table 2: The presentation of “Density” and “Dielectric Constant” data for Al2O3 (20-30 and 50) is confusing.
Other issues that could be included in the work are:
- Nothing is said about the amount of nano fluids (NF) that is prepared each time, nor about their price. Is the use of NF in electrical power transformers viable?
- In the work the authors talk about long-tern stability. Fluids work for years in power transformers. How long did it take from the manufacturing of the NF to the testing of their stability?
- In power transformers, NFs work in conditions of temperature different from ambient and under a strong magnetic field. Would these conditions affect the results offered at work?
Author Response
Work presentation is a little confusing. The introduction comments: “… influence of conducting (Fe3O4), semi-conductive (ZnO), and insulating (ZrO2, SiO2, and Al2O3) nanoparticles …”
But the order in which the results are presented does not follow this order of NPs; moreover, in different sections it follows different orders.
- It would be appreciated if the results of the different tests of the NPs were always presented following the same order of NPs.
Authors’ response
This comment has been taken into account.
Authors’ Action
The results of the different tests of the NPs are presented following the same order of NPs.
- In some parts of the work (for example: Fig 4) the NPs sizes are included next to the name of the NPs, while in others the sizes are not included; The presentation of the work should be uniform.
Authors’ response
This comment has been taken into account.
Authors’ Action
The NPs sizes are now indicated next to the name of NPs.
- Other aspect is that the work has an excessive number of tables and figures, those that provide redundant information should be eliminated, for example, Table 4 does not provide new information to that is presented in Figures 5 and 6; it is recommended to delete the table, or delete both figures.
Authors’ response
This comment has been taken into account.
Authors’ Action
Tables 4 and 5 have been removed.
- Table 1: “Density at 20 (°C)”, density units are missed.
Authors’ response
This has been corrected (Tables 1 and 2).
Authors’ Action
Density units are added.
- Table 2: The presentation of “Density” and “Dielectric Constant” data for Al2O3 (20-30 and 50) is confusing.
Authors’ response
This has been corrected. Table 2 is clearer now.
Authors’ Action
The indications of “Density” and “Dielectric Constant” data for Al2O3 (20-30 and 50) are presented in the appropriate columns.
Other issues that could be included in the work are:
- Nothing is said about the amount of nanofluids (NF) that is prepared each time, nor about their price. Is the use of NF in electrical power transformers viable?
Authors’ response
The amount of prepared nanofluids is 400 mL for each concentration and type of nanoparticles. Indeed, we did not analyze the cost of the NFs (price of the base liquid, nanoparticles, preparation) nor the comparison with the base liquid price. As concerns the viability of NFs for power transformers, it's too early to pronounce.
Authors’ Action
The following sentence has been added (subsection 2.2):
“Note that the amount of prepared nanofluids is 400 mL for each concentration and type of nanoparticles.”
- In the work the authors talk about long-tern stability. Fluids work for years in power transformers. How long did it take from the manufacturing of the NF to the testing of their stability?
Authors’ response
The stability of the tested NFs was checked one week after their preparation. Note that after three weeks we didn’t observe sedimentation. However, if the zeta potential indicates that NFs remains stable, this cannot be a guarantee of stability for several years and for the lifetime of a transformer. A study should be carried out in this direction to ensure the stability of NFs and to possibly find solutions to this problem.
- In power transformers, NFs work in conditions of temperature different from ambient and under a strong magnetic field. Would these conditions affect the results offered at work?
Authors’ response
We didn’t investigate the influence of temperature nor that of magnetic field on the behavior of NFs. Thus, it’s is difficult to speak out at this stage.
It should be emphasized that the interest in nanofluids (NFs) and their possible use in high voltage (HV) applications was initially motivated mainly by their well-known property that is their interesting heat transfer. However, the increase in temperature should improve the dielectric strength of the nanofluid in a first phase until the moisture is eliminated and then, when the temperature will be very high and the nanofluid begins to pass into vapor phase, the dielectric strength should decrease.
This being, one can think that the magnetic field would affect the homogeneity of nanofluids with magnetic nanoparticles (Fe2O3) and thence the breakdown voltage.
Note that in power transformers using the insulating fluid for cooling, fluid flowing over pressboards can generate static electricity which can be hazardous when the level of ECT (Electrostatic Charging Tendency) reaches a critical value. Research on this topic is ongoing in our laboratory.
Reviewer 2 Report
I am familiar with many of A. Beroual's papers. This paper is very laborious and extensive, and also confirms the high level of technical research. However, several questions remain.
1. The main, and in my opinion, unsolved problem of nanofluids is sedimentation over time. In the article, this issue is considered only indirectly, through the measurement of the zeta potential. I would like to confirm experimentally the weak influence (or absence on influence) of the nanofluid storage time on the dielectric strength by direct measurement.
2. Any effect on dielectric strength is due to the effect on the initiation or propagation of the discharge. During initiation, the question remains unclear, from which electrode and at what polarity, the discharge is ignited. The explanations of the authors, in my opinion, are insufficient. The capture of electrons during the ignition of a discharge from the cathode can both prevent ignition and promote it. If the discharge is ignited at the anode, then it is completely incomprehensible what the capture of electrons has to do with it. In my opinion, it is necessary to take into account, for example, the effect of electrical double layers both near the nanoparticles and at the electrodes.
3. The paper is too large, it is difficult to read such a long text. It should be shortened. Extensive arrays of data are duplicated both in graphs and in tables.
4. The mechanism by which PD occurs remains unclear.
Author Response
Comments and Suggestions for Authors
I am familiar with many of A. Beroual' s papers. This paper is very laborious and extensive, and also confirms the high level of technical research. However, several questions remain.
- The main, and in my opinion, unsolved problem of nanofluids is sedimentation over time. In the article, this issue is considered only indirectly, through the measurement of the zeta potential. I would like to confirm experimentally the weak influence (or absence on influence) of the nanofluid storage time on the dielectric strength by direct measurement.
Authors’ response
Indeed, the problem of sedimentation (homogeneity) and long-term stability over several years or even decades (in the order of the service life of transformers) remains unsolved and requires long-term studies.
The stability of the tested NFs was checked one week after their preparation. Note that after three weeks we didn’t observe sedimentation. However, if the zeta potential indicates that NFs remains stable, this cannot be a guarantee of stability for several years and for the lifetime of a transformer. A study should be carried out in this direction to ensure the stability of NFs and to possibly find solutions to this problem.
Authors’ Action
The following sentence has been added in subsection 3.1:
Note that after three weeks we did not observe sedimentation. However, if the zeta potential indicates that NFs remains stable, this cannot be a guarantee of stability for several years and for the lifetime of transformer.
- Any effect on dielectric strength is due to the effect on the initiation or propagation of the discharge. During initiation, the question remains unclear, from which electrode and at what polarity, the discharge is ignited. The explanations of the authors, in my opinion, are insufficient. The capture of electrons during the ignition of a discharge from the cathode can both prevent ignition and promote it. If the discharge is ignited at the anode, then it is completely incomprehensible what the capture of electrons has to do with it. In my opinion, it is necessary to take into account, for example, the effect of electrical double layers both near the nanoparticles and at the electrodes.
Authors’ response
This work focuses on the dielectric strength of nanofluids. The tests are therefore carried out according to IEC 60156 standard method, in a quasi-uniform electric field. The phenomena of initiation and propagation of streamers are therefore not the subject of this work. Of course, in the case of a divergent field, the development of the streamers is a function of the polarity of the electrodes and the voltage waveform.
Note that the mechanism of the electrical double layer (EDL) has been discussed in the paper (see penultimate paragraph of section 4 (Discussion).
- The paper is too large; it is difficult to read such a long text. It should be shortened. Extensive arrays of data are duplicated both in graphs and in tables.
Authors’ response
This comment has been taken into account.
Authors’ Action
Tables or figures that are redundant are removed.
- The mechanism by which PD occurs remains unclear.
Authors’ response
Partial discharges’ (PDS) mechanism is complex. One thing is certain, it is that the appearance of PDs is linked to a strengthening of the electric field on a given site (contamination, particles in suspension, cavities (gas bubble or water droplets), protrusion, defect ...). In the case of NFs, as the base liquid is pure (degassed and filtered), it is the nanoparticles that will play on the initiation and development of PDs. Depending on the nature of the NPs and their interactions with electrons, which will be essential.
Authors’ Action
The following paragraph has been added (last paragraph – section 4 Discussion)
Depending on the properties of the NPs and the liquid used, the presence of NPs can induce electric field heterogeneity in the NF. The localized increase of the electric field can lead to PDs at the same applied voltage or at the same ionization level of the liquid. Thus, near the electrodes, high mobility electrons and low mobility ionized ions are ready to migrate under the electric field forces. The NPs then play the role of electrons and negative ion scavengers, which generate a potential well (trapping site) that reduces the electron/ negative ion movement and thence PD activities [10,36]. When the trapping process is finished, nanoparticle surfaces are saturated with negative charges; hence they no longer could trap more electrons. The limit is strongly correlated with the mismatch between base liquid and NP's conductivities and/or permittivities [10,35].